# A composite subunit vaccine confers full protection against Buruli ulcer disease in the mouse footpad model of *Mycobacterium ulcerans* infection

Justice Kofi Boakye-Appiah[1,2], Andy C. Tran[1], Matthew J. Paul[1], Peter Hart[1], Richard O. Phillips[2], Thomas S. Harrison[1], Mark Wansbrough-Jones[1], Rajko Reljic[1]*

**1** Institute for Infection and Immunity, School of Health and Medical Sciences, City St George's University of London, London, United Kingdom, **2** School of Medical Sciences, Kwame Nkrumah University of Science and Technology (KNUST), Kumasi, Ghana

\* rreljic@sgul.ac.uk

## Abstract

Buruli ulcer (BU) disease, a neglected necrotizing tropical skin infection caused by *Mycobacterium ulcerans*, is the third most common mycobacterial disease after tuberculosis and leprosy. Infections mostly occur in remote, rural areas of Central and West Africa, but also in Australia, Japan and Papua New Guinea. There is currently no vaccine against Buruli ulcer disease and all previous attempts using closely related bacteria and subunit proteins have been partially successful only. Here, we tested in mice a composite subunit formulation incorporating the *Mycobacterium ulcerans* toxin mycolactone as the immunomodulator, and the antigens Ag85A and Polyketide Synthase Enzyme Ketoreductase A (KRA), formulated with Quil-A adjuvant ('Burulivac'). Burulivac induced Ag85A and KRA antigen-specific antibodies, T cells and a mixed pro- and anti-inflammatory cytokine responses, which conferred absolute protection against Buruli ulcer disease in the mouse footpad model over a 14-week period of observation. This was superior to both live attenuated mycobacterial vaccines, that is, BCG and an avirulent *M. ulcerans* strain that lacks the mycolactone toxin (*Mu*Δ). Interleukin 10 was found to be strongly associated with protection. We suggest that Burulivac is a promising vaccine candidate against Buruli ulcer disease that warrants further exploration.

## Author summary

Buruli ulcer is a neglected tropical disease caused by skin infections by *Mycobacterium ulcerans*, an organism related to causative pathogens of tuberculosis and leprosy. The disease is endemic to parts of Central and West Africa but is also found in Australia, Japan and some other countries. While mortality is low, morbidity caused by extensive and unsightly ulceration of the skin is high. There currently is no vaccine against Buruli ulcer and all previous attempts to develop one have been unsuccessful.

**Data availability statement:** All relevant data are within the manuscript, its Supporting information files and minimal datasets accessible through https://doi.org/10.6084/m9.figshare.28280519 and https://doi.org/10.6084/m9.figshare.28280516.

**Funding:** This work was supported by the Horizon 2020 work programme grant 643558 to RR, with contributions from St George's Hospital Charity award (to MWJ), UK Medical Research Council award (to ROP) and VALIDATE Network award (to JKB-A). The funders had no role in study design, data collection and analysis, decision to publish, or preparation of the manuscript.

**Competing interests:** We have read the journal's policy, and the authors of this manuscript have the following competing interests: JKB-A is currently employed by Eli Lily and Co, and PH is currently employed by the CEPI organisation. All other authors declare that no conflicting interests exist.

Here we developed a vaccine formulation termed 'Burulivac' that incorporates three *M. ulcerans* antigens, two of which are proteins and the third the mycolactone toxin, much responsible for the pathology of the BU disease. Our rationale was that in a vaccine setting and using a controlled dose of the toxin, we could calibrate the immune responses to selected antigens, so that they mimic those induced during natural infection. Using an experimental mouse footpad model of infection, we demonstrate that immunisation with Burulivac completely prevented ulceration in the mouse footpad over an extended 14-weeks period of observation, and that this correlated with elevated levels of anti-inflammatory cytokine Interleukin-10. We propose that Burulivac is a promising vaccine candidate that merits further testing and development.

## Introduction

*Mycobacterium ulcerans (Mu)* infection causes a panniculitis with extensive necrosis of the subcutaneous fatty tissue which leads to the formation of ulcers with undermined edges known as Buruli ulcer (BU). It is recognized as a common and serious disease in some parts of the tropics [1]. In recent times, outbreaks of Buruli ulcer disease in various countries including Australia have brought the infection into the limelight, with data suggesting it to be the third most common mycobacterial disease after tuberculosis and leprosy. The mode of transmission is not established but the prevailing view is that water insect bites in Africa [2] and mosquitos in Australia [3] may be responsible. BU disease presents as a painless skin lesion and may be of one of five forms: nodule, papule, plaque or oedematous lesion and then progressively ulceration [4]. Ulcers are the most commonly identified forms especially in places with poor healthcare.

*Mu* produces a lipid-like toxin molecule called mycolactone (ML), first isolated and characterised in 1998 [5,6], which is much responsible for disease pathogenesis and pathology. It comes in variant structures with differences even though the central components of the structure remain the same. They are similar in structure to macrolides produced as secondary metabolites by soil bacteria [7], such as *Streptomyces* and *Saccharopolyspora* species in the order *Actinomycetales*. George et al. demonstrated that mycolactone was the main pathological factor involved in the disease process [6] since inoculation of guinea pigs with ML deficient strains of *Mu* did not cause the characteristic BU lesions. However, site injections of purified ML alone resulted in characteristic BU lesions [8,9]. The mechanisms of ML pathogenicity are multiple but the critical for its cytotoxicity and immunomodulatory activity is targeting of the Sec61 protein secretion pathway in host cells [10,11]. There is currently no published evidence whether ML is a target for immune responses or whether it could play any role in vaccine formulations. The current dogma is that mycolactone is immunosuppressive, inhibiting multiple T-cell pathways and the expression of cytokines and chemokines [12].

There currently exists no vaccine against Buruli. Vaccine studies against BU date back to 1956 when Fenner carried out immunizations with BCG, low or high dose *Mu*, and *Mycobacterium balnei* (marinum). He concluded that BCG protection was poor and even though *M. marinum* and *Mu* provided some limited protection against BU, this protection was not by antibody transfer [13,14]. Similarly, Fraga et al [15] in an attempt to develop a vaccine devoid of the immunosuppressive toxin ML, tested an ML-deficient strain of *Mu* (M51, or *MuΔ*, [16]) due to repeated sub-culturing, leading to the spontaneous loss of MUP038 sequence encoding some genes that are involved in the synthesis of ML [17]. In this study, the strain only managed to delay onset of swelling post-challenge but was unable to protect against it

altogether. Even though BCG and *MuΔ* have proven to only confer temporary protection, these animal experiments have provided important lessons on immunological determinants of protection and immunological correlates involved in BU immunity. For example, experiments demonstrated the interplay of cytokines during the protective phase and the period during which mice developed footpad swellings on challenge. These principles and lessons have been adopted and applied in designing and trying various vaccine candidates against BU in the mouse model of infection [18]. These mouse trials involving several antigenic candidates [19,20] have demonstrated partial protection at best [21]. This suggests that protein antigens alone may not be sufficient or do not induce immunologically relevant response in absence of immunomodulation by ML. It also may suggest that ML itself, while a virulence factor in infection, may also be required for protection.

In this study, we tested in the mouse footpad model of *Mu* infection a composite vaccine formulation containing for the first time the ML toxin itself, alongside the Polyketide Synthase Enzyme KR A (KRA) involved in its synthesis, and the cell wall mycolic acid transferase Ag85A, formulated with the Quil-A adjuvant (Burulivac formulation). We analysed immune responses induced by Burulivac in mice, in comparison with *MuΔ* and BCG live attenuated mycobacterial strains and determined the effect of vaccination on clinical and microbiological outcomes, by measuring the foot lesions over a 14-week period and the bacterial content in affected tissue. We found Burulivac to be a highly effective vaccine, completely preventing foot ulceration in mice.

## Materials and methods

### Ethics statement

All mice used in this study were used according to the UK national legislation (Animals in Scientific Procedures Act, 1986) under the UK Home Office animal project license number P1A7411AD. The ethical approval was obtained from Animal Welfare and Ethical Review Body (AWERB) of St George's University of London, under Ref No ID3F0AC7E-2017.

### Mycolactone and recombinant proteins

ML was obtained in 0.5 ml glass vials from Yoshito Kishi's laboratory at Harvard University. Samples came in ethyl acetate or ethanol carrier media at concentrations of 200 μg/ml. This ML is WHO verified and has been used extensively by researchers in other labs for BU related work. KRA and Ag85A antigens, alongside Acyltransferase propionate (ATP), Acyltransferase acetate 2(ATAC 2) and Enoyl Reductase (ER), were expressed from plasmids obtained from Tim Stinear's laboratory at the University of Melbourne. These were expressed from transformed BL21 Plys *E. coli* strain (Invitrogen) induced by 1 mM Isopropyl β-D-1-thiogalactopyranoside (IPTG). Protein purification of His-tagged denatured proteins reconstituted in urea from inclusion bodies was done using a Sepharose-Nickel affinity column (GE 17-0575-01 Chelating Sepharose Fast Flow). The eluted proteins were then dialysed in PBS and endotoxin removed by running the protein solution through Polymyxin B agarose column (Sigma-Aldrich), which was verified by performing a LAL test (Thermofisher Scientific). Proteins were then concentrated (Amicon), aliquoted and stored at −20°C. Standard Coomassie and Western blot protocols (detecting His tag) were performed to confirm the presence and purity of the proteins.

### Coomassie staining and Western blotting of recombinant proteins

Coomassie and Western blots were performed to further characterise and confirm the presence of purified proteins. This also aided in estimating the percentage purity of the proteins

post-purification. The protein samples were incubated with LDS loading buffer (Bio-Rad) and β-mecaptoethanol (Sigma-Aldrich) at 85ºC for 5 min and immediately cooled on ice for 2 min. Loading was then done onto 4–12% w/v BIS-TRIS SDS-PAGE gels (Invitrogen) alongside a reference ladder (Biorad Precision Plus Protein, All Blue Standards) and run at 140V for 1 h in MES buffer (Life Technologies, Novex). The proteins in the gel were then either stained with Coomassie dye or transferred onto a nitrocellulose membrane for a Western blot using the semi-dry transfer method (0.04A per gel for 60 minutes). Nitrocellulose membranes were blocked with 5% w/v skimmed milk protein (Marvel original) in PBS overnight at 4ºC and then incubated with monoclonal anti-polyhistidine–peroxidase antibody (Sigma-Aldrich) for 2 h at 1:1,000 dilution in PBS 5% w/v milk. The membrane was then washed between staining steps three times with TBS 0.1% w/v Tween-20. To visualize the proteins, the ECL peroxidase substrate (Amersham) was added and the image captured on a gel reader (Syngene G box).

## Mouse immunisations

C57BL6 female mice (age 10–12 weeks) purchased from Harlan were used in the experiments. Nine mice were used in each experimental group, with two to three used for immunological readouts and six for *Mu* infection. Numbers of experimental mice are indicated in figure legends. All immunisations were carried out via the subcutaneous route. With ML being a fat-soluble molecule, the subcutaneous route presents an opportunity to inject it into the fatty layer of the skin, mimicking the likely route of human infection. Mice immobilised in a rodent restrainer were immunised subcutaneously at the base of the tail. Each mouse received 100 μl of various vaccine formulations as detailed in Table 1. In prime-boost regimens, primed mice were re-immunised after three weeks rest, followed by another boost after a further three weeks. Two weeks after the final immunisation, mice were culled for immunogenicity studies or infected with *Mu*. Mice were monitored for at least 30 minutes after each injection to ensure they did not develop any acute adverse reactions such as skin irritations, swellings, bleeding, paralysis etc. The various vaccine components in formulations were as follows (per dose): antigens, 10 μg; ML, 0.5 μg; Quil-A (Sigma), 15 μg; BCG and *MuΔ*, $10^5$ colony-forming units (CFU). Both BCG (Pasteur) and *MuΔ*(M51), as well as virulent S141 *Mu* strain used subsequently in pathogenic challenge studies, were retrieved from St George's long-term storage repository of mycobacterial strains.

**Table 1. immunisation regimens.**

| Regimen | Priming (1x) | Boosting (2X) 3-weekly periods |
|---|---|---|
| *Arm A* | *Complete vaccine formulations* | |
| 1 | PBS | PBS |
| 2 | BCG | PBS |
| 3 | *MuΔ* | PBS |
| 4 | *MuΔ* | Burulivac |
| 5 | Burulivac | Burulivac |
| *Arm B* | *Vaccine components* | |
| 6 | *MuΔ* | Ag85A+Quil-A |
| 7 | *MuΔ* | KRA+Quil-A |
| 8 | *MuΔ* | ML+Quil-A |

## Mouse infection/challenge with *Mu*

Infection with wild type S141 strain of *Mu* was done by directly inoculating the right foot-pads of mice. First, the mice were anaesthetised using isoflurane in gas chambers supplied with oxygen. Once fully anaesthetised, animal was placed on tissue towel on the back. Using flat-end clean forceps to stabilise the right foot, a preloaded syringe containing approximately $10^5$ organisms in 0.05 ml PBS is then used to inject inoculum into the middle of the mouse footpad, at a 45° angle. Injections were carried out slowly to prevent backflow. When properly angulated and well timed, there was no resistance to flow. Success of the inoculation was confirmed visually: the footpad became visibly inflated and there was no backflow. The uninoculated left footpad was used as a control. Each animal was then observed in the cage until it regained conciseness and normal mobility.

Prior to culling mice, footpads were measured in experiments which involved footpad infections. On the presumption that mouse footpads are of the same height, dimensions of footpad breadth and thickness are taken using an electronic digital vernier calliper (Visenta). These measurements were taken weekly. To keep mice still during measurements, they were anaesthetised with inhaled isoflurane in a gas chamber.

## Tissue processing

For serum collection after culling the mice, abdominal dissections were performed, and blood samples obtained from the intraperitoneal space by cutting into the abdominal aorta and collecting on average 1.0 ml of free-flowing blood. Vials containing collected blood were then left overnight at room temperature to clot and for serum to separate. Clotted blood was then centrifuged at a speed of 15,000 g for 30 minutes and separated serum was later collected and stored at −20° C for subsequent antibody ELISA assays.

Spleens harvested from immunised mice were transported in RPMI medium (supplemented with 100 μg/ml Penicillin, 100 μg/ml streptomycin, 10% bovine serum albumin and 0.3 g/L of Glutamine) on ice. The spleens were disrupted by passing the tissue through 70 μm cell strainers using a syringe plunger, and then incubated for 5 minutes in ACK lysis buffer (Sigma-Aldrich) to lyse the red blood cells. The splenocytes were then washed twice with the RPMI medium to remove debris and seeded into round bottom 96-well tissue culture plate wells at concentrations of 500,000 cells per well for various assays. Counting and viability were performed by trypan blue (ThermoFisher) staining method.

After culling mice, footpads were obtained by amputating the foot at the ankle joint. Amputated foot pieces were immediately placed in 4 ml homogeniser tubes containing 1.5 ml of PBS and ceramic homogeniser beads (Part of the Precellys Evolution homogenizer lysing kit). Amputated footpads were disrupted by using a homogenizer (Precellys Evolution from Bertin Technologies) at a speed of 7,500 Revolutions per Minute and in three cycles lasting 25 seconds with 90 second breaks between cycles. Using cut tip pipettes, liquid components of the resulting homogenate were collected and stored frozen in liquid nitrogen. Some of this was later used for plating at various dilutions and *Mu* DNA quantification, while the remaining homogenate was filtered through 0.45 μm filters. The resulting filtered supernatant was used in C-reactive Protein assays (ThermoFisher Mouse CRP ELISA Kit).

## *Mu* culture and enumeration in footpad tissues

Prior to culture of *Mu*, the modified Petroff homogenization and decontamination method was performed to remove other bacteria and fungi. Homogenates were placed into 50 ml Falcon tubes and an equal volume of 4% NaOH was added and tubes shaken for 15 minutes on a shaker, before centrifugation was carried out at 3,000 rpm for 15 minutes. The

supernatant was discarded and sterile 0.9% NaCl added to the pellet, which was centrifuged again as before. Afterwards the supernatant was discarded, and pellet resuspended in 0.9% NaCl before growing bacteria. Middlebrook 7H9 media supplemented with ADC (Albumin Dextrose Catalase) growth supplement was used for liquid culture, and Middlebrook 7H11 supplemented with OADC (oleic acid + ADC) for plating. Contamination was reduced by supplementing the medium with a mixture of the antimicrobials (PANTA, [22]) prior to inoculation. PANTA contains polymyxin B, amphotericin B, nalidixic acid, trimethoprim, and azlocillin. *Mu* cultures (liquid or plates) were grown in 30°C incubator with 5% $CO_2$ atmosphere for 4–14 weeks, with weekly checks for growth and possible contamination.

### Antigen recall assays and flow cytometry

Spleens obtained from culled immunised mice were homogenised and seeded into tissue culture plates (in triplicates of 500,000 splenocytes per well). The wells contained 200 μl of RPMI media supplemented with 10%FBS and L-glutamine. 5 μg of corresponding antigens were added to the wells and appropriate controls were set up. After 3 days of incubation, the samples were centrifuged at 400 g and supernatant harvested for various cytokine ELISAs while the pelleted cells were washed for onward flow cytometric analysis.

Splenocytes were washed in 200 ul of PBS three times and re-suspended in 25 ul per well of viability dye (Invitrogen). The plate was then incubated at 4°C for 20 minutes. Following this, 200 ul of FACS buffer (consisting of PBS, 0.5% Bovine Serum Albumin and 0.1% Sodium Azide) was added to each well. The plate was then spun at 400 g for 5 minutes and supernatant discarded, leaving cells at the bottom of the plate. A mastermix of antibodies was prepared and 25 ul of this was added to each well. Control wells with a master mix minus one antibody were also setup for each antibody used. The plate was incubated at 4°C for another hour and cells washed with 200 ul per well afterwards. Cells were prepared for flowcytometry according to the Biolegend kit procedure for staining cells in u-bottom plate wells. The following antibody-colour panel was used. Viability Dye: Efluor 780/APC-Cy7, CD3+: AF488, CD4+: PerCP-Cy5.5, CD8+: BV510, CD45+: RA PE/Cy7, CD69+: PE, Ki67: APC. 10,000 cells were acquired on the CytoFlex (Beckman Coulter) and analysed using FlowJo V10 software. CD3 positive cells were gated for either CD4+/Ki67+ or CD8+/Ki67+ phenotype (proliferating cells) and data expressed as a proportion of total CD4+ or CD8+ cells.

### Cytokine detection assays

In another objective of this set of experiments and following three days of incubation of spleen cells with corresponding *Mu* antigens, the plates were centrifuged at 400 g and supernatant harvested for various cytokine ELISAs (ThermoFisher Scientific) including IFN-γ, TNF-α, IL-10 and IL-17A, IL-6 and IL-2, using the manufacturer's instructions.

### Antibody detection

ELISA plates (Maxisorb, Thermo Fisher Scientific) were coated with 5 μg of corresponding proteins in 100 μl of PBS per well overnight at 4°C. The wells were then washed thrice with 200 μl of 0.5% PBS Tween-20 per well. The wells were then blocked with 200 μl 5%w/v PBS skimmed milk for 2 h at room temperature. Washing was again performed (3x). Corresponding sera samples serially diluted in the blocking buffer were then added and the plate sealed and incubated overnight at 4°C. After repeating the washing step, wells were incubated with a secondary conjugated antibody (Sigma-Aldrich) at 1:1,000 dilution in 5% W/V PBS skimmed milk for 1 h at room temperature and then washed again (5x). Colour signal was

then developed using SigmaFast OPD substrate. The developing signal was read at 450 nm by a plate reader (Tecan Infinite 200 Pro).

### *Mu* DNA quantification

A *Mu*-specific DNA quantification method was employed to measure residual bacterial load adapted from the method of Beissner et al., 2012 [23]. DNA was extracted according to the Qiagen Standard Operating Procedure (SOP) for DNA Extraction with QIAamp MINI Kit and qPCR performed to detect IS2404 of *M ulcerans*. For the qPCR, the following primers were used in the preparation of the reaction mix; Primer IS*2404* TF1: 5' aaa gca cca cgc agc atc t′3 (TibMolBiol, Berlin, Germany), Primer IS*2404* TR1: 5' agc gac ccc agt gga ttg′3 (TibMolBiol), Probe IS*2404* TP2: 'FAM-ccg tcc aac gcg atc ggc a-BBQ′3 (TibMolBiol), 5x HOT FIREPol Probe qPCR Mix Plus (no ROX), 1 ml (SolisBioDyne, Tartu, Estonia, ref# 08- 15-00001). PCR was run at 40 cycles, with 15 seconds at 95°C denaturation and 60 seconds at 60°C annealing/extension cycles. To exclude contaminations with *M. ulcerans* DNA, No Reactive template controls (containing H2O instead of template), a No template control and negative extraction controls (IS*2404* qPCR) were processed in the same way as the samples.

### CRP assay

An ELISA-based assay for murine CRP-protein was used in accordance with the manufacturer's instructions (Abcam, ab157712). Briefly, supernatant from homogenised mouse footpads was harvested for this assay. 50 μL was added to wells in the ELISA plate and 50 μL of the manufacturer's antibody cocktail added to the corresponding wells. Following incubation at room temperature for an hour, each well was washed three times with 1x of the manufacturer's wash buffer and 100 μL TMB development solution was added to each well for an additional 20 min incubation at room temperature. 100 μL of stop solution was then added to the wells and plate was read for optical density at 450 nm.

### Statistics

Statistical analyses were performed using GraphPad Prism 7 and Microsoft Excel 2013. One-way ANOVA multi-comparison test followed by Tukey's post hoc correction were performed to determine statistically significant differences between various conditions, as defined by $p \leq 0.05$. Confidence levels are indicated by a single or multiple asterisks, as indicated in figures. Experiments were carried out in duplicates or triplicates (stated in figure legends). Bars represent averages of biological or technical repeats and error bars represent standard deviations of the mean. Specific circumstances are described in detail in the results and discussion sections.

## Results

### Generation of recombinant *Mu* proteins

*Mu* antigens Ag85A, KRA, ATP, ATAC2 and ER were expressed in *E. coli* and purified to a high degree by chelating Nickel chromatography, but subsequent selection of Ag85A and KRA for inclusion in the Burulivac vaccine formulation was based on preliminary immunogenicity studies of humoral and cellular responses induced by these antigens, as described later. Following removal of endotoxin and reconstitution of denatured antigens in PBS, the proteins were analysed by SDS-PAGE (Fig 1A) and Western blotting (Fig 1B) using anti-His antibodies. Ag85A displayed an apparent Mw of 37 kDa, which is marginally larger than 35 kDa reported previously [24], while KRA showed an Mw of 25 kDa, as reported previously [25].

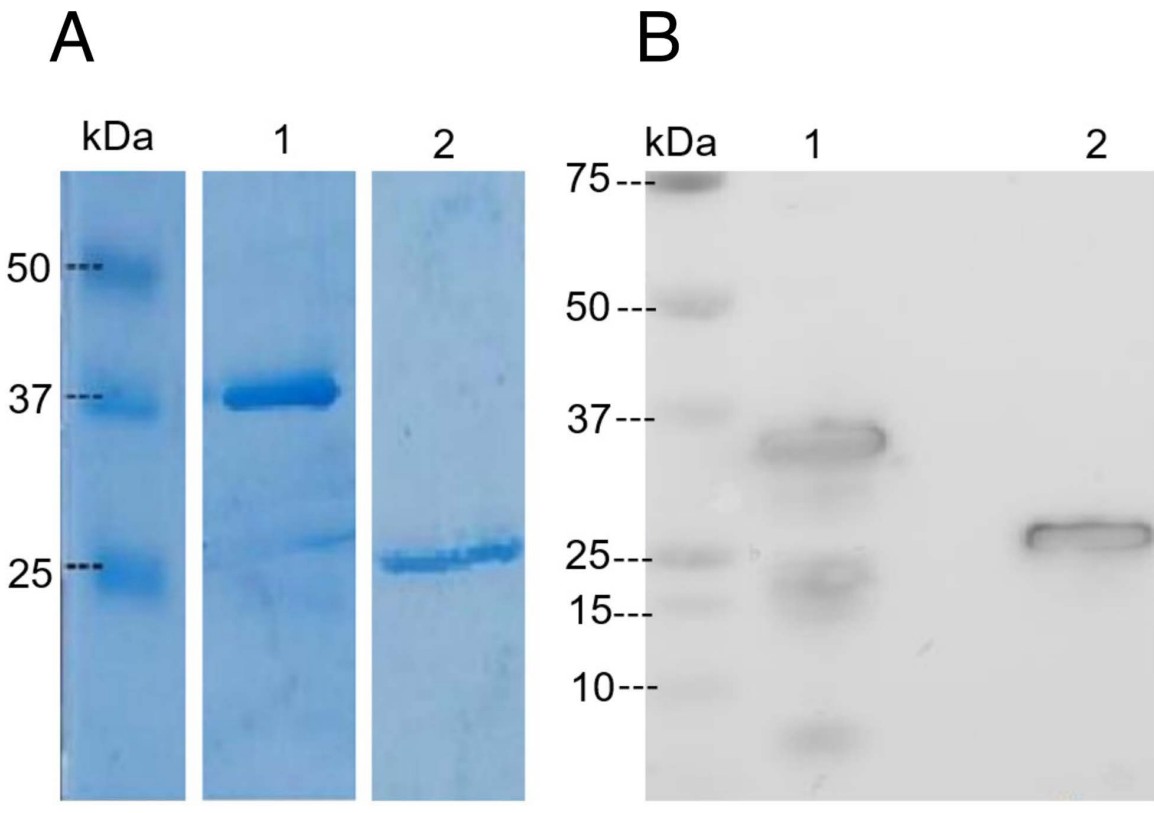

**Fig 1. SDS-PAGE and Western blot detection of recombinant proteins.** 5 μg of each protein is loaded onto 10% polyacrylamide gel and following separation, detected by Coomassie Blue staining (A) or anti-His antibody following nitrocellulose transfer (B). Lane 1: Ag85A (main protein bend detected at 37 kDa on Coomassie, or 35 kDa on Western); Lane 2: KRA (protein band visible at 25 kDa).

## Immunogenicity of *Mu* antigens in mice

Initial immunisation experiments were performed to find out if the selected antigens could induce immune responses in mice. Following two subcutaneous immunisations with *Mu* antigens formulated in Quil-A adjuvant, we tested for antigen specific IgG responses in sera and found that Ag85A and KRA induced highest IgG antibody response (Figs 2A and S1). We then tested for cellular responses in antigen restimulated spleen cell cultures by measuring cytokine secretion in response to antigenic recall and for T cell proliferation. Ag85A induced strong IL-17 and IFNγ responses, whereas KRA induced IFNγ and only moderate levels of IL-17A (Fig 2B and 2C). These responses were superior to the three other antigens tested, ATP, ATAC2 and ER (S2 and S3 Figs), which led us to further investigate responses to Ag85A and KRA only. In an antigen recall cultures assay, we could observe proliferation of CD4 and CD8 T cells to Ag85A but not KRA (Fig 2D and 2E). Taken together, these data showed that both Ag85A and KRA could induce antibodies, but Ag85A appeared to be the more immunogenic in inducing T cell immunity, while the other three protein antigens were less immunogenic and were not further investigated. We attempted to investigate potential immune responses specific to ML, but this proved challenging, since it is a non-proteinaceous molecule and there are currently no reliable protocols to detect them. Thus, in a pilot study of its potential immunogenicity, we tested for capacity of ML to induce antibody and cellular responses, when formulated with different adjuvants and delivery systems. The 500-ng dose was selected as it was not cytotoxic *in vitro*, either in J774 macrophage cells or mouse whole splenocyte cultures

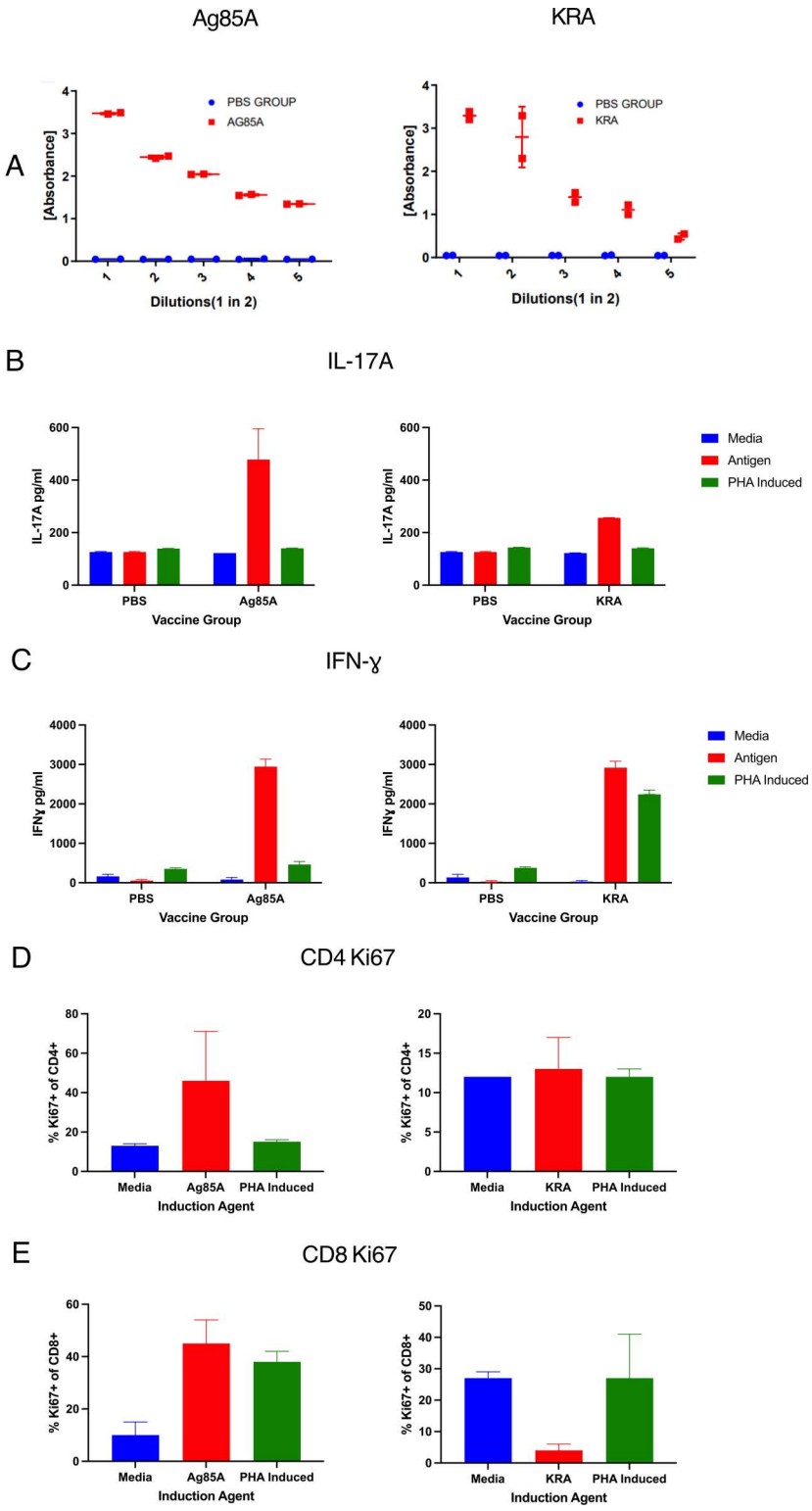

**Fig 2. Immunogenicity of *Mu* antigens Ag85A and KRA in mice.** A) IgG responses in sera following two subcu-taneous immunisations (shown are 2-fold serial dilutions; N = 2 mice). B) IL-17 detected in supernatant of spleen cultures following antigen restimulation for 3 days. C) IFN-γ responses in the same supernatants; D and E) CD4 and CD8 T cell proliferation after 3-day restimulation of spleen cells with respective antigens or PHA. N = 3; shown are the means, with error bars indicating SD.

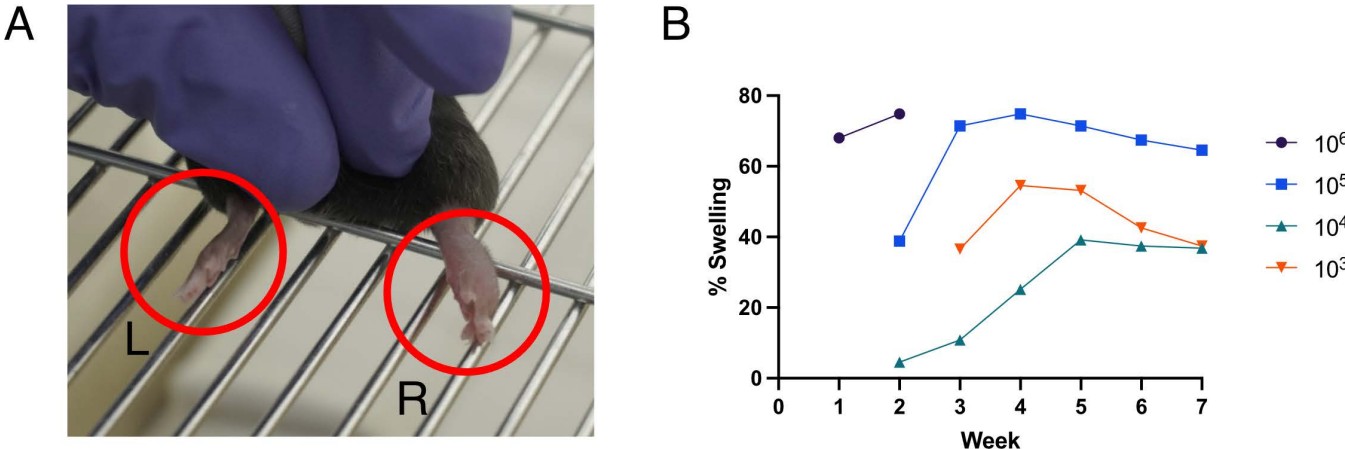

**Fig 3. Mouse model of *Mu* footpad infection.** A) Swelling in the right footpad of a mouse in the $10^6$ *Mu* bacteria group at week 1. Left footpad used as a control showed no change. B) Trend curves of average percentage change in right footpad size measured weekly from point of infection for seven successive weeks. Values are means from 5–6 mice.

(S4 Fig). We failed to detect ML-specific antibody responses in sera but splenocytes of mice immunised with ML formulated with Quil-A and no other adjuvants displayed a proliferative response upon restimulation with ML (S5 Fig). Interestingly, this response was observed predominantly in non-T cell compartment, while we observed only a low-level proliferation in conventional (α/β) and none in unconventional (γ/δ) T cell compartments (S5 Fig). Thus, based on these immunogenicity studies, we concluded that Ag85A and KRA were the most immunogenic protein antigens, and that ML, in addition to its immunomodulatory role may also be a target itself for the immune system. Therefore, we opted to include it in the final vaccine formulation (Burulivac) based on both these properties.

### The mouse footpad model of *Mu* infection

Having confirmed immunogenicity of the selected antigens, we then proceeded with testing the final vaccine formulation which alongside Ag85A and KRA antigens also included ML, all formulated in Quil-A adjuvant and termed 'Burulivac' vaccine candidate. To do this, we first needed to establish a reliable animal model of BU disease to test its efficacy. The mouse footpad model [26] is by far the most used model in preclinical studies of *Mu* infection. We injected one hindleg footpad with different numbers of *Mu* bacilli whereas the other footpad served as the negative control for onset of BU lesions. Depending on the infectious dose, the infected footpads developed visible lesions at various timepoints which could be measured by diameter using a digital vernier calliper from Visenta (Fig 3A). The highest dose ($10^6$ bacilli) induced the lesions fastest, within the first couple of weeks and these mice reached pathology endpoints prior to ulceration stage. The size and dynamics of lesions growth in other mouse groups were positively associated with inoculation dose, although $10^3$ and $10^4$ doses did not follow this trend. (Fig 3B). Overall, we opted for the $10^5$ doses as they gave the most reliable increase in lesions size over the 7-week experimental period.

### Immunogenicity of *Burulivac* and vaccine components

Mice were immunised with different vaccine formulations as listed in Table 1 and according to the protocols and timelines described in the methodology. Arm A of the experiment included complete vaccine formulations, while arm B included the three individual antigenic

components. Immune responses in vaccinated mice were tested only for arm A vaccine regimens but the protection following *Mu* challenge was measured for all vaccine groups.

On recall with Ag85A (Fig 4A), splenocytes from all immunised groups in Arm A except the PBS control, secreted very high levels of IFN-γ, reaching the upper detection limit of the test kit used. Ag85A responses in the Burulivac groups were directed to IL-2, IL-10 and TNF-α, whereas IL-6 responses were muted. Interestingly, the two Burulivac groups gave the highest IL-10 responses, whether *MuΔ* was used to prime or not. As for the two attenuated vaccines, BCG and *MuΔ*, they both induced elevated levels IL-2, TNF-α and IFN-γcompared to the control group, but with exception of IFN-γ, these were inferior to the two Burulivac groups. The cytokine profile after KRA stimulation (Fig 4B) followed a similar trend, except for the BCG group failing to induce any cytokines, as KRA is not present in the BCG proteome.

## Footpad swelling after *Mu* challenge

We used percentage change in footpad volume as a proxy to determine severity of infection and degree of protection conferred by the respective complete vaccines (arm A). Graphs of percentage change in footpad size were plotted for comparison to control footpad (Fig 5). The largest swelling was observed in the PBS group with a percentage change in infected footpad volume of 46%. BCG vaccinated animals displayed a percentage change in volume of 38%, followed by *MuΔ* group which recorded average percentage change in footpad size of 10%. Strikingly, the Burulivac groups, whether with *MuΔ*prime or not, both displayed no measurable swelling at all (Fig 5A). Thus, although *MuΔ*alone was highly protective in our experiments, the Burulivac alone group provided similar or better protection, making the priming with *MuΔ*redundant.

In arm B of the experiment, individual antigenic components were tested for their contribution to protection against foot swelling. Each component, i.e., Ag85A, KRA or Mycolactone, was used as a boost to *MuΔ*, and each was formulated with Quil-A adjuvant. PBS and *MuΔ* alone from the arm A of the experiment were used as the reference points, as the two arms of the experiment ran concurrently. The results (Fig 5B) revealed that each of the three Burulivac vaccine components used to boost *MuΔ* conferred near complete protection, which was however statistically no different to *MuΔ* alone, implying that individually they did not contribute to protection conferred by *MuΔ*.Yet, when combined together in the triple formulation, they were protective even in absence of *MuΔ*, implying that there was a distinct additive or synergistic effect at work, which translated into complete protection. Due to the complexity of the experimental designs, we were unable to test additional combinations, for example combinations of two rather than three vaccine components, so we cannot exclude possible redundancies.

## Clinical outcomes in terms of disease *vs* no disease

A binary endpoint of footpad swelling was investigated as a measure of protection from disease. To do this, mice within each vaccine group from arm A of the experiment were categorized based on footpad size as to whether they were fully protected or not protected. Any form of swelling was considered not protected. The number of protected mice were then represented as percentage of the total (Fig 5C). All mice in the PBS group developed swelling, therefore, there was 0% protection. The BCG group recorded 20% protection with 80% of mice in the group developing swellings of varying degrees. *MuΔ* conferred absolute protection to 70% of the mice vaccinated. Both Burulivac groups whether primed with *MuΔ* or not conferred absolute protection from swelling in all mice (100%).

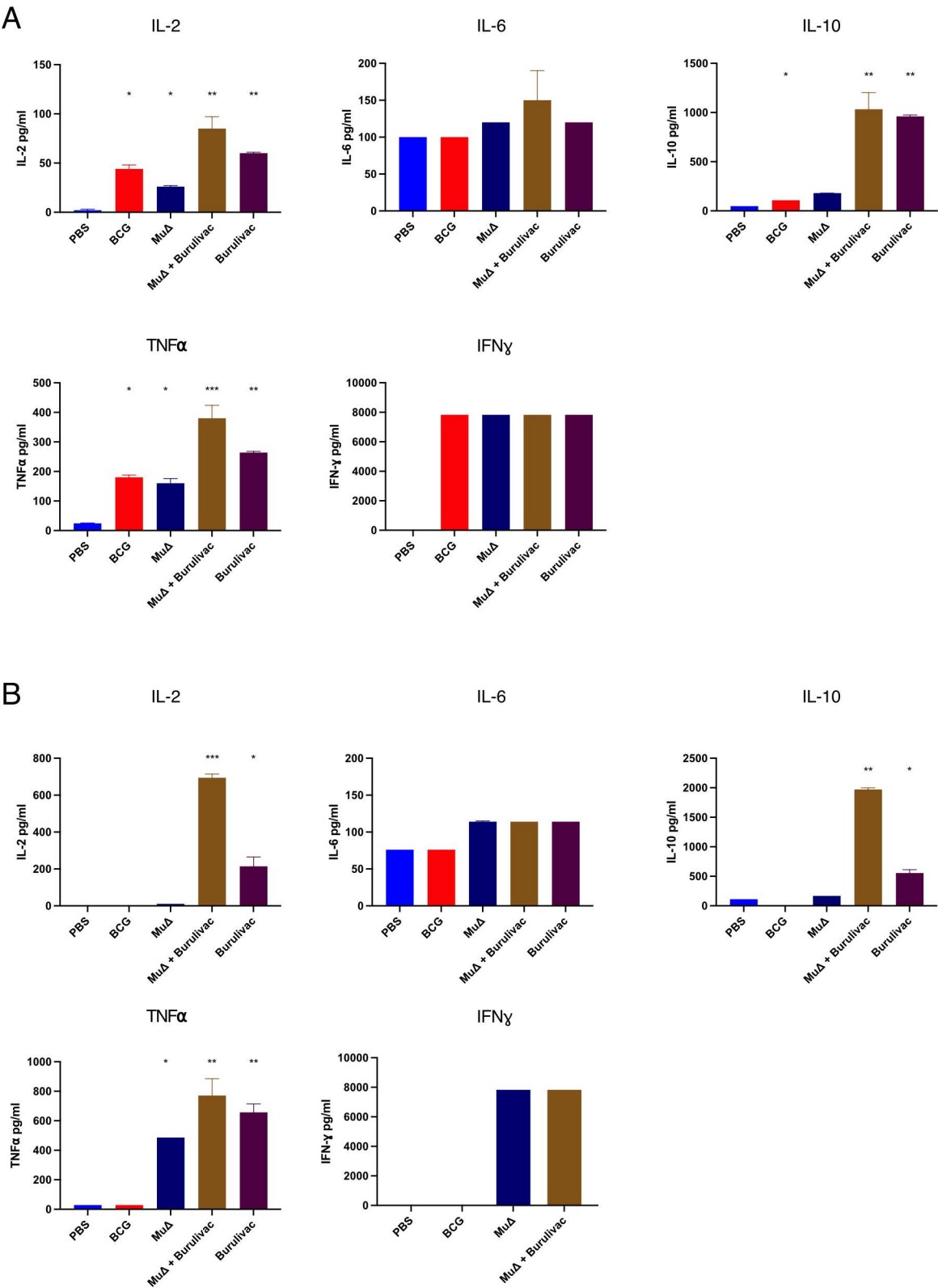

**Fig 4. Cytokine profile of splenocytes of immunised mice stimulated with Ag85A or KRA.** Following immunisation as indicated in Table 1 Arm A, three mice from each group were culled before Mu challenge, and cytokine responses evaluated in spleen cultures stimulated with Ag85A (A) or KRA (B). Measured cytokines were IL-2, IL-6, IL-10, IFN-γ and TNF-α. Comparison performed by 1-way Anova and post hoc Tukey test. N = 3, with * indicating < 0.5, **< 0.01 and *** < 0.005 and error bars indicating SD.

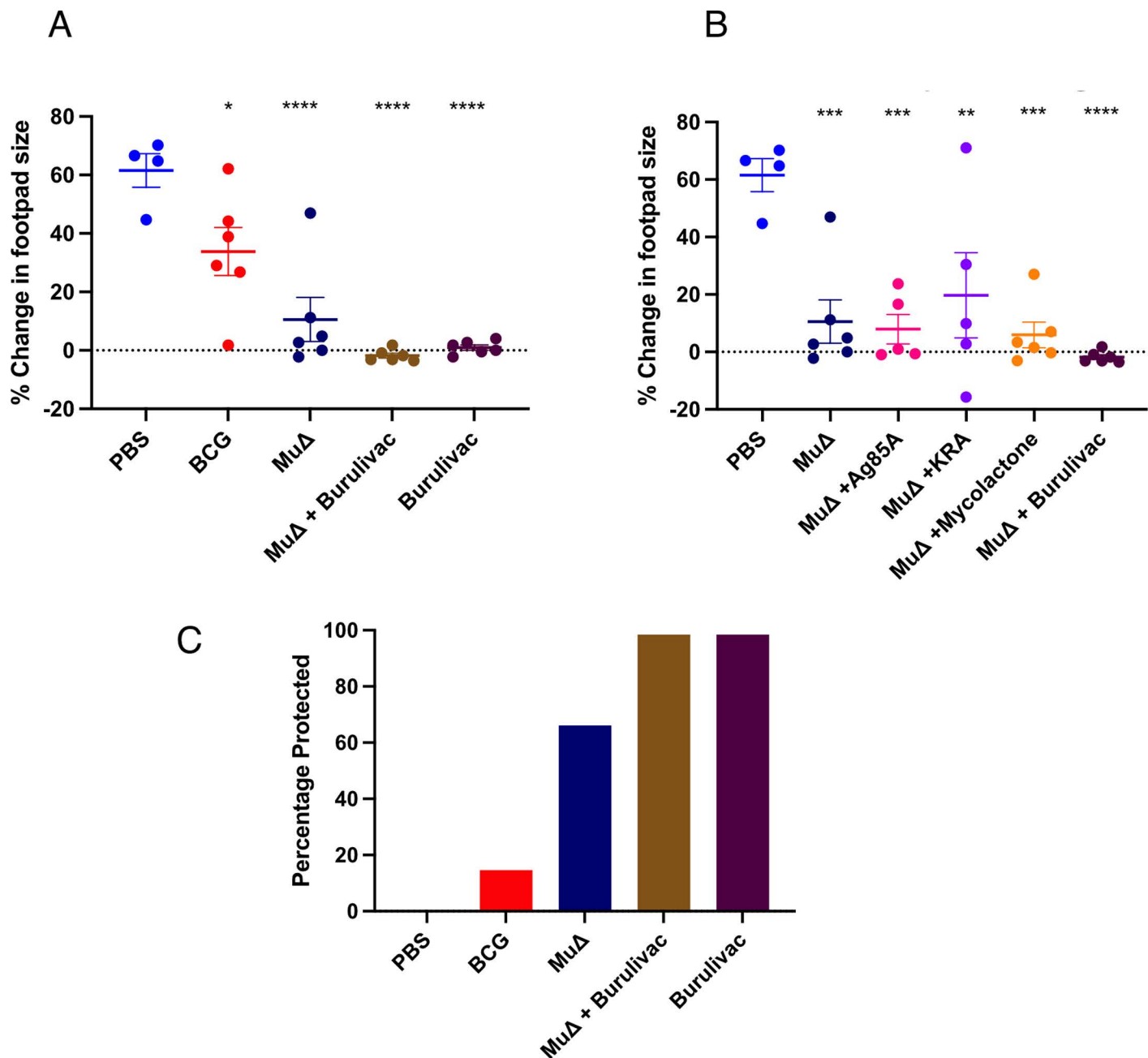

**Fig 5. Percentage change in footpad swelling (size) in infected mice immunised with (A) complete formulations or (B) individual vaccine components.** Right mouse footpads were infected with $10^5$ *Mu* bacteria while the left footpad was used as a control. Percentage change in swelling was calculated relative to the control footpad. Comparison performed by 1-way Anova and post hoc Tukey test. N = 5–6, with error bars indicating SD. Significance: * < 0.5, ** < 0.01, *** < 0.006 and **** < 0.001. C) Percentage protection of mice within specific groups, e.g., mice which did not develop any swelling at all. Swelling was determined by both physical observation and measurement with callipers. Animals with any observed or measured swelling were classified as 'not protected'.

## Residual *Mu* DNA copies and CRP in infected footpads

Combined 16S rRNA RT/IS2404 specific for *Mu* assays were performed on homogenates of infected mouse footpads (Fig 6A). This was to determine the remnant DNA count in the footpads at the time of cull. There was a wide variation in the population in each group.

Expectedly, the number of DNA copies was highest in the control PBS group. The Burulivac groups contained no DNA material, as did the *MuΔ* only group, apart from one mouse. DNA counts in the BCG group were low and statistically different from both PBS and Burulivac groups. It is however remarkable to note that the DNA counts were very significantly higher in the PBS group than in all the vaccine groups.

To determine the extent of inflammation ongoing in the infected pads, CRP concentration in the supernatant of each homogenised amputated foot was determined by ELISA CRP kit. C-Reactive Protein (CRP) is an acute phase reactant produced by the liver in response to inflammatory injury. It has both pro and anti-inflammatory properties but is used as a marker of inflammation [27]. The data show that PBS group displayed the highest levels of CRP, while fully protected groups (i.e., Burulivac regimen) showed the lowest levels of CRP (Fig 6B). All other groups (which were partially protected) showed variable but generally low levels of CRP. Thus, the observed protection in Burulivac regimen is associated with absence of an excessive inflammatory response in the infected footpads.

## Discussion

In this study, we demonstrated that a vaccine formulation consisting of *Mu* antigens Ag85A, KRA, and the mycolactone toxin, formulated with Quil-A adjuvant, (Burulivac) conferred complete protection in the mouse footpad model of *Mu* infection. Burulivac was equally protective as a homologous prime-boost regimen, or when used to boost *MuΔ* strain vaccine, an avirulent form of *M. ulcerans*.

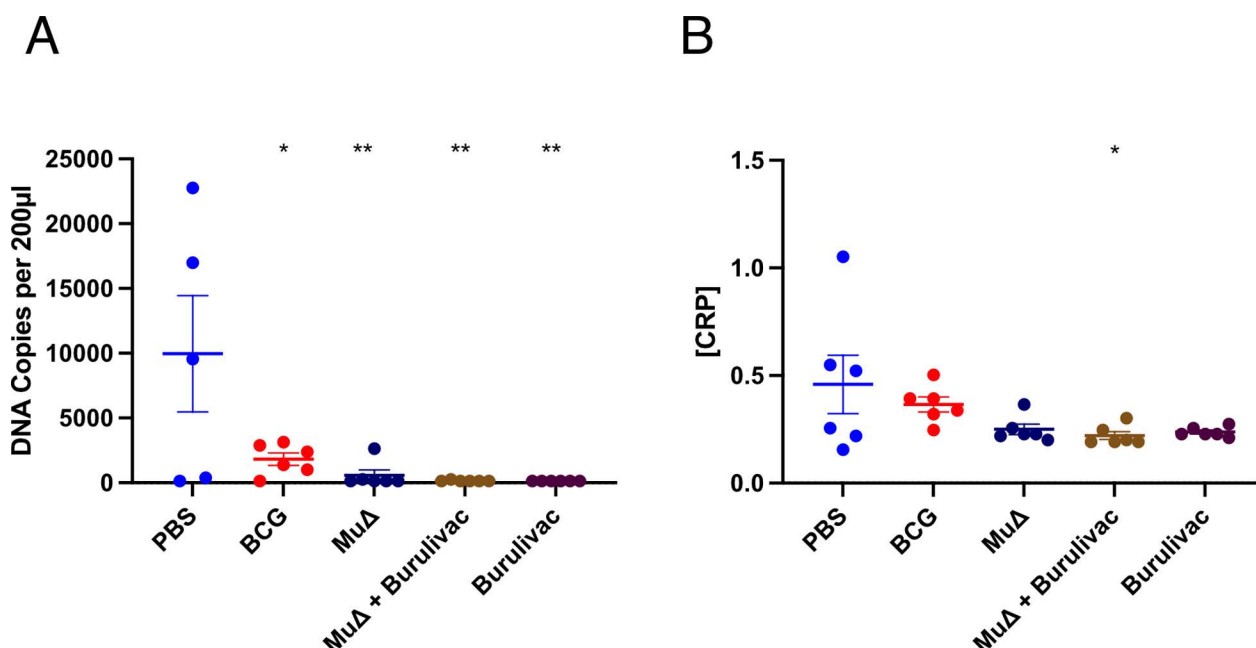

**Fig 6. Residual *Mu* DNA copies and CRP in infected footpads.** A) Combined *Mu* 16S rRNA RT/IS2404 DNA copy count in infected mouse footpads at the point of cull. Vaccinated mice were challenged with Log4 *Mu* bacteria and monitored for weeks until the pre-determined experimental end point for each mouse. *Mu* DNA copy counts were taken at the end point. B) C reactive protein concentrations in infected footpads as determined by ELISA method using ThermoFisher CRP ELISA kit. Optical density was measured at 492 nm. N = 5–6 (an extra mouse in each group was culled without measurement of swelling, for technical protocols validation, but tissues could be included here). Comparison performed by 1-way Anova and post hoc Tukey test, with error bars indicating SD. Significance: * < 0.5, **< 0.01.

The ML deficient *Mu* strain 5,134 (*MuΔ*), was found to be more protective than reported previously [15]. While *MuΔ* clearly does have potential as a vaccine candidate for BU, it may need to be combined with other antigens to achieve complete protection. Here, we investigated several approaches to boosting MuΔ with other antigens. MuΔ is genetically highly homologous (98%) to virulent strains of *Mu*, and therefore retains most of the antigens of the wild type bacteria but without the secretion of the immunosuppressive ML [16]. It is expected to present many common antigens without the immunomodulation observed in infection. We observed here that *MuΔ* conferred greater protection against swelling than BCG, confirming previous work demonstrating that even though BCG can confer some protection, this is only moderate and transient [28,29].

*Mu* specific Ag85A is expressed as a cell membrane antigen in *MuΔ* [30]. Ag85A recall responses were therefore expected in all vaccine groups, including BCG which has its own homologue of this antigen. Thus, except for the PBS control, all other vaccine groups gave very strong IFN-γ responses. However, IL-6 responses could not be detected in any of the vaccine groups. Remarkably though, IL-2, TNF-α and especially IL-10 responses were the highest in Burulivac vaccine groups after Ag85A re-stimulation of splenocytes. In contrast to cell membrane associated Ag85A, Keto-reductase A (KRA) is a cytosolic enzyme module involved in the synthesis of ML by multi-domain PKS enzymes [31]. As a cytosolic antigen, immune responses to KRA are expected to be limited during natural infection, but such vaccine-induced 'unnatural immunity' may potentially prove effective. Indeed this was shown for the cholera vaccine, with vaccine induced antibodies correlating better with protection than those induced by natural exposure [32]. KRA was indeed found to be immunogenic as demonstrated by both antibody and cytokine responses, and so, could be potentially a protective antigen against BU disease. This is consistent with previous work demonstrating that a KRA-based DNA vaccine induced partial protection in mice [17].

Our main rationale for inclusion of ML in our vaccine formulations was to calibrate immune responses induced by protein antigens against those induced in natural infection settings, so that they might be relevant to protection. To that effect, the ML dose of 500 ng in our formulations was chosen empirically, bearing in mind the fact that average ML levels measured from tissue samples taken from infected patient lesions do not exceed 500 ng/ml [33].

Ultimately, the extent of protection against BU disease conferred by various vaccine candidates was measured. This was done by looking at two parameters. The first was to determine if the vaccine candidates played any role in the reduction of footpad swelling and thus severity of disease suffered. With the PBS group setting the baseline average percentage change in footpad size at 46%, any change in footpad size below this was considered protection. All vaccine candidates thus mitigated in the severity of disease suffered by the mice. However, Burulivac, the vaccine candidate comprising Ag85A, KRA and ML combined with Quil A turned out to give the best protection. There was no footpad swelling in any of the mice vaccinated with this candidate. One group was primed with *MuΔ* while the other was not. There was however no difference in the level of protection observed in the two groups, with both showing complete protection. Even though pro-inflammatory cytokine responses were highest in Burulivac groups, the IL-10 response was also the highest in these two groups among all the vaccine groups. IL-10 as previously discussed is anti-inflammatory in action and is in this case seen to correlate with protection. Our observation therefore goes to reinforce the ambivalent interplay of pro and anti-inflammatory factors involved in protection against BU disease. It also affirms the observation of Mangas et al [18] who concluded following mouse experiments that a potent vaccine against BU disease must induce tissue-specific immune profiles with controlled inflammatory responses at the site of infection. From our data, we add that these responses should include high titres of cytokines such as IL-10. Further credence is given to this by the

observation of low titres of CRP as opposed to the higher titres of IL-10 in footpads of mice with the least footpad swelling.

When the number of protected mice with no footpad swelling at all is represented as percentage of the total, the clinical effect of Burulivac vaccine becomes more apparent. All mice in the PBS group developed swelling. There was therefore 0% protection. The BCG group recorded 20% protection with 80% of mice in the group developing swellings of varying degrees. Without any boost, *MuΔ* prime conferred absolute protection to 70% of the mice vaccinated. In contrast, both Burulivac groups conferred absolute protection in all mice (100%), suggesting that the combination of the three antigens is sufficient for protection, while boosting *MuΔ* with individual antigens was not. This was also corroborated by complete absence of *Mu* DNA copies in footpad tissues in the Burulivac groups.

A weakness of our study is that due to logistical constraints, we could not test all possible antigenic combinations or vaccine regimens to ensure that there is no redundancy. This remains to be established in future studies, as indeed does the precise role of ML in the observed protection. While we hypothesize that it is an immunomodulatory one, it is also possible that ML could itself be a target for the immune response in some form, though there is currently no evidence for such in literature. We also failed to detect antibody responses to ML in our study (not shown), though we observed some cellular proliferation. However, its inclusion in our vaccine formulation in combination with the two protein antigens may have played a role in conferring the observed complete protection against BU disease in the mouse footpad model of infection, since individual antigens could not do so. We therefore conclude that the Burulivac vaccine formulation is sufficient and probably necessary for this potent protection and that this is associated with a delicate balance of pro- and anti-inflammatory mechanisms that may have been driven by ML. Guided by these findings, we intend to progress with Burulivac in future studies to determine the longevity of protection offered while delving deeper into its mechanisms and correlates of protection.

It should also be noted that in terms of future translation of this experimental vaccine candidate into human application, we foresee two main challenges. The first is that a formulation that contains ML may be perceived as unsafe, which may be an obstacle in obtaining regulatory approvals for human clinical trials. We argue though that at a fixed low dose as applied in this study, and in absence of live *Mu* that would continue to generate it, ML is very unlikely to cause localised tissue pathology. Indeed, in our mouse experiments we did not observe one, in any of mice inoculated with the vaccine dose of ML. Progressive swelling was observed only in animals that were pathogenically challenged with live *Mu*. Also, although ML alone induced local tissue apoptosis in the guinea pig skin model, this required a dose of 0.1 mg [8], which is 200-fold higher than what we used in our vaccine study in mice. It is therefore reasonable to argue that there is a window of immunomodulatory effect without significant cytotoxicity, but exploring this in detail was beyond the scope of the current study. However, this would have to be tested in human safety clinical trials, most likely in a dose escalating manner till reaching the intended vaccine dose.

The second and probably more significant challenge concerns vaccine development and potential licensure. Being a relatively minor infectious disease in terms of at-risk population, BU is not an attractive target to industry for vaccine development. Vaccine development and clinical testing typically cost several hundreds of millions of US$, and such investment would be extremely difficult to recoup in case of BU, even over a long period of time. This is also the reason why the current WHO policy on BU is heavily focused on improving case detection and treatment, rather than driving vaccine development. Nevertheless, being a preventable infectious disease, this should not stop the scientific drive to

develop a vaccine and therefore, our study contributes to the ongoing efforts to tackle this neglected tropical disease, with the view that it may be easier to change policies, political will and commercial priorities once a promising vaccine candidate exists, rather than in absence of it. Our findings suggest that a protective BU vaccine is an attenable target, and as such, warrants further consideration.

## Supporting information

**S1 Fig. IgG Antibody ELISA responses in sera of mice immunised with various *Mu* antigens of interest.** Shown are two-fold dilutions of sera starting from 1:100, in comparison to PBS immunised mice. The best titrating responses was in the Ag85A and KRA vaccinated groups while $ATAC_2$ gave the poorest response. N = 2.
(TIFF)

**S2 Fig. *In vitro* Interferon Gamma secretion by splenocytes of various antigenic mouse groups upon recall with corresponding antigens.** The colours indicate what the recall material was.i.e., Blank media (Negative control), Antigen of interest/immunisation and the Phytohemagglutinin (PHA) which was used as a positive control. Error bars represent standard deviation of the mean and bars are means of triplicate wells. N = 2.
(TIFF)

**S3 Fig. *In vitro* Interleukin-17 Interferon gamma secretion by splenocytes of various antigenic mouse groups upon recall with corresponding antigens.** The colours indicate what the recall material was.i.e., Blank media (Negative control), Antigen of interest/immunisation and the phytohemagglutinin (PHA) which was used as a positive control. Error bars represent standard deviation of the mean and bars are means of triplicate wells. N = 2.
(TIFF)

**S4 Fig. Cytotoxic effect of Mycolactone on cells.** Cells after 72 h of incubation with different concentrations of mycolactone were incubated with Resazurin for 4–6 hours to determine survival. There is a direct relation between cell viability and absorbance (Excitation wavelength of 540 nm and Emission wavelength of 580 nm). J774 and mouse spleen cells maintained high survival rates. Error bars represent Standard Deviation from triplicate measurements.
(TIFF)

**S5 Fig. Flow cytometric determination of cell type populations of mycolactone-immunised mice upon recall with mycolactone.** Bars are colour-coded by recall antigen. APC stands for antigen-presenting cells; these were enriched by splenocyte adherence to culture flask, pre-treated with mycolactone and then combined with whole splenocytes). Anti-CD3 antibody was used as a positive control for T cell proliferation. Efluor 780 was used to monitor cell viability, while-Ki67 was used as a marker for proliferation. Proliferating cells are expressed as percentage of total cells. Error bars indicate Standard Deviation from triplicate measurements. Combined splenocytes from 2 mice.
(TIFF)

## Acknowledgment

We would like to thank Prof Tim Stinear (University of Melbourne, Australia) for provision of plasmids expressing *M. ulcerans* antigens, Prof Kris Huygen (formerly of Institute Pasteur, Brussels) for helpful advice, Prof Yoshito Kishi (formerly of Harvard University, MA, US) for provision of mycolactone and Dr Rachel Simmonds (University of Surrey, UK) for help with mycolactone cytotoxicity assays.

## Author contributions

**Conceptualization:** Justice Kofi Boakye-Appiah, Peter Hart, Richard O. Phillips, Thomas S. Harrison, Mark Wansbrough-Jones, Rajko Reljic.

**Data curation:** Andy C. Tran, Matthew J. Paul, Richard O. Phillips, Rajko Reljic.

**Formal analysis:** Justice Kofi Boakye-Appiah, Rajko Reljic.

**Funding acquisition:** Justice Kofi Boakye-Appiah, Richard O. Phillips, Mark Wansbrough-Jones, Rajko Reljic.

**Investigation:** Justice Kofi Boakye-Appiah, Rajko Reljic.

**Methodology:** Justice Kofi Boakye-Appiah, Andy C. Tran, Matthew J. Paul, Peter Hart, Rajko Reljic.

**Project administration:** Rajko Reljic.

**Resources:** Mark Wansbrough-Jones.

**Supervision:** Rajko Reljic.

**Writing – original draft:** Rajko Reljic.

**Writing – review & editing:** Justice Kofi Boakye-Appiah, Andy C. Tran, Matthew J. Paul, Richard O. Phillips.

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
