## [Decision Letter · Decision Letter 0]

10 Jan 2025

PNTD-D-24-01702

A composite subunit vaccine confers full protection against Buruli ulcer disease in the mouse footpad model of Mycobacterium ulcerans infection

Dear Dr. Reljic,

Thank you for submitting your manuscript to PLOS Neglected Tropical Diseases. After careful consideration, we feel that it has merit but does not fully meet PLOS Neglected Tropical Diseases's publication criteria as it currently stands. Therefore, we invite you to submit a revised version of the manuscript that addresses the points raised during the review process.

Please submit your revised manuscript within 60 days Mar 11 2025 11:59PM. If you will need more time than this to complete your revisions, please reply to this message or contact the journal office at plosntds@plos.org. Please include the following items when submitting your revised manuscript:

We look forward to receiving your revised manuscript.

Kind regards,

Elsio A Wunder Jr, DVM, Ph.D.

Section Editor

Elsio Wunder Jr

Section Editor

Shaden Kamhawi

co-Editor-in-Chief

Paul Brindley

co-Editor-in-Chief

**Additional Editor Comments (if provided):**

**Journal Requirements:**

**Reviewers' Comments:**

Reviewer's Responses to Questions

**Key Review Criteria Required for Acceptance?**

**Methods**

-Are the objectives of the study clearly articulated with a clear testable hypothesis stated?

-Is the study design appropriate to address the stated objectives?

-Is the population clearly described and appropriate for the hypothesis being tested?

-Is the sample size sufficient to ensure adequate power to address the hypothesis being tested?

-Were correct statistical analysis used to support conclusions?

-Are there concerns about ethical or regulatory requirements being met?

Reviewer #1: Objectives are clear and inline with the hypothesis stated. The study design is appropriate to address the objectives. To enhance reproducibility, provide more detail on the specific assays used to measure immune response.

There are no ethical concerns.

Reviewer #2: (No Response)

Reviewer #3: The objectives of the study was well articulate and thoroughly described.

The study is appropriate for the set objectives. The number of mice in each group of immunisation arm is not clearly indicated but can be only inferred from Fig. 5 as 6 or 5 mice per arm. This has to be clearly indicated in the method section.

**Results**

-Does the analysis presented match the analysis plan?

-Are the results clearly and completely presented?

-Are the figures (Tables, Images) of sufficient quality for clarity?

Reviewer #1: The analyses presented are robust.

Reviewer #2: (No Response)

Reviewer #3: The results are clearly and appropriate presented with figures and images.

**Conclusions**

-Are the conclusions supported by the data presented?

-Are the limitations of analysis clearly described?

-Do the authors discuss how these data can be helpful to advance our understanding of the topic under study?

-Is public health relevance addressed?

Reviewer #1: All conclusions are supported by the results.

Reviewer #2: (No Response)

Reviewer #3: The conclusions are supported by the data presented and study limitations are also described. Overall the study set the stage for further investigation into the vaccine candidate for Buruli ulcer.

**Editorial and Data Presentation Modifications?**

Reviewer #1: (No Response)

Reviewer #2: (No Response)

Reviewer #3: (No Response)

**Summary and General Comments**

Reviewer #1: The very well written manuscript presents a significant contribution in the field of vaccine development against BU as the results are promising. The use of the composite sub-unit formulation is innovatives and relevant in the fight against BU.

The study design and use of the mouth foodpad model, is apropriate and results are compelling, demonstrating absolute protection in the treated groups.

The authors could address the potential challenges in translating these findings to human applications.

Reviewer #2: The manuscript by Boakye-Appiah et al. presents the use of a composite subunit vaccine formulation, named "Burulivac", in the mouse footpad model of Mycobacterium ulcerans infection. While the development of a vaccine for Buruli ulcer (BU) is an important area of neglected diseases research and the observed prevention of footpad ulceration in mice challenged with M. ulcerans after receiving Burulivac is interesting, the proposed vaccine strategy and evaluation of data have several significant shortcomings:

1. In the presented vaccine formulation, mycolactone, which is responsible for much of the pathology of BU, is included in its native, toxic form. What is the reasoning for using the native toxin rather than a modified, attenuated version? Several effective toxoid vaccines, such as those for diphtheria, tetanus, or botulism rely on inactivated forms of the toxin. To develop a vaccine that is safe for humans, stimulating an immune response without causing harm, it is essential to use an attenuated form of the toxin. Using the native, toxic form therefore does not appear to be a viable approach for developing a human vaccine for Buruli ulcer. Therefore, I do not agree with the authors’ main conclusion that Burulivac is a promising vaccine candidate for BU.

2. The authors state that they included mycolactone as the immune modulator but do not provide evidence for this role of mycolactone in their manuscript. While they assess immune responses in vaccinated mice against the two included protein antigens, they do not analyze responses against mycolactone. They mention that they did not directly test for any potential immune responses specific to mycolactone, stating that “it is a non-proteinaceous molecule and there are currently no established protocols to detect them”. However, other research groups have reported the development and use of ELISAs, which can determine antibody responses to mycolactone. Analyzing antibody responses to mycolactone after vaccination could provide valuable insights for improving future vaccine design. Considering that antibodies are crucial for anti-toxin immunity, antibody responses to mycolactone may be the defining factor in the observed protection. If this is not the case, it would be crucially important to characterize the proposed immune modulatory activity of mycolactone, as other groups have reported that strong immune responses against a range of M. ulcerans proteins confer no or only limited protective efficacy in the mouse footpad model.

3. The authors do not provide a clear rationale for their choice of the composite subunit vaccine formulation. It would be helpful if they could explain why they selected and combined Ag85A, KRA, and mycolactone. Furthermore, no dose-finding experiments have been reported to determine the optimal concentrations of these components. This is particularly important for mycolactone, to provide insight into its contribution to vaccine efficacy.

4. What is the rationale behind the choice of the different immunization regimens used in this study? Why were mice in Arm A groups 1 to 3 immunized only once with BCG and Mu delta, while groups 4 and 5 were immunized 3 times (according to Table 1)? This discrepancy makes a valid comparison of the performance of vaccine formulations to prevent footpad ulceration across the groups in Arm A impossible. It is well-established that boosting leads to more effective immune responses, so this variation in regimen complicates the interpretation of results. Essentially, there is no valid comparative regimen to the proposed Burulivac regimen. Group 4 was first immunized with Mu delta and then boosted with Burulivac so even Mu delta vaccination cannot be directly compared to Burulivac. Therefore the performance of Burulivac compared to all other regimens remains unclear.

Reviewer #3: (No Response)

PLOS authors have the option to publish the peer review history of their article (what does this mean? ). If published, this will include your full peer review and any attached files.

**Do you want your identity to be public for this peer review?** For information about this choice, including consent withdrawal, please see our Privacy Policy .

Reviewer #1: No

Reviewer #2: No

Reviewer #3: **Yes: ** Michael Frimpong

**Figure resubmission:**
---

## [Decision Letter · Decision Letter 1]

2 Feb 2025

Dear Dr. Reljic,

We are pleased to inform you that your manuscript 'A composite subunit vaccine confers full protection against Buruli ulcer disease in the mouse footpad model of Mycobacterium ulcerans infection' has been provisionally accepted for publication in PLOS Neglected Tropical Diseases.

Best regards,

Elsio A Wunder Jr, DVM, Ph.D.

Section Editor

Elsio Wunder Jr

Section Editor

Shaden Kamhawi

co-Editor-in-Chief

Paul Brindley

co-Editor-in-Chief

Reviewer's Responses to Questions

**Key Review Criteria Required for Acceptance?**

**Methods**

-Are the objectives of the study clearly articulated with a clear testable hypothesis stated?

-Is the study design appropriate to address the stated objectives?

-Is the population clearly described and appropriate for the hypothesis being tested?

-Is the sample size sufficient to ensure adequate power to address the hypothesis being tested?

-Were correct statistical analysis used to support conclusions?

-Are there concerns about ethical or regulatory requirements being met?

Reviewer #3: The authors have addressed all comments raised previously by reviewer and updated the manuscript accordingly.

**Results**

-Does the analysis presented match the analysis plan?

-Are the results clearly and completely presented?

-Are the figures (Tables, Images) of sufficient quality for clarity?

Reviewer #3: (No Response)

**Conclusions**

-Are the conclusions supported by the data presented?

-Are the limitations of analysis clearly described?

-Do the authors discuss how these data can be helpful to advance our understanding of the topic under study?

-Is public health relevance addressed?

Reviewer #3: (No Response)

**Editorial and Data Presentation Modifications?**

Reviewer #3: (No Response)

**Summary and General Comments**

Reviewer #3: (No Response)

PLOS authors have the option to publish the peer review history of their article (what does this mean? ). If published, this will include your full peer review and any attached files.

**Do you want your identity to be public for this peer review?** For information about this choice, including consent withdrawal, please see our Privacy Policy .

Reviewer #3: **Yes: ** Dr. Michael Frimpong

---

## [Editor Report · Acceptance letter]

Dear Dr. Reljic,

We are delighted to inform you that your manuscript, "A composite subunit vaccine confers full protection against Buruli ulcer disease in the mouse footpad model of Mycobacterium ulcerans infection," has been formally accepted for publication in PLOS Neglected Tropical Diseases.

Best regards,

Shaden Kamhawi

co-Editor-in-Chief

Paul Brindley

co-Editor-in-Chief
